# New Rules for Domain Independent Lifted MAP Inference

**Happy Mittal, Prasoon Goyal**
Dept. of Comp. Sci. & Engg.
I.I.T. Delhi, Hauz Khas
New Delhi, 110016, India
happy.mittal@cse.iitd.ac.in
prasoongoyal13@gmail.com

**Vibhav Gogate**
Dept. of Comp. Sci.
Univ. of Texas Dallas
Richardson, TX 75080, USA
vgogate@hlt.utdallas.edu

**Parag Singla**
Dept. of Comp. Sci. & Engg.
I.I.T. Delhi, Hauz Khas
New Delhi, 110016, India
parags@cse.iitd.ac.in

## Abstract

Lifted inference algorithms for probabilistic first-order logic frameworks such as Markov logic networks (MLNs) have received significant attention in recent years. These algorithms use so called *lifting rules* to identify symmetries in the first-order representation and reduce the inference problem over a large probabilistic model to an inference problem over a much smaller model. In this paper, we present two new lifting rules, which enable fast MAP inference in a large class of MLNs. Our first rule uses the concept of *single occurrence equivalence class of logical variables*, which we define in the paper. The rule states that the MAP assignment over an MLN can be recovered from a much smaller MLN, in which each logical variable in each single occurrence equivalence class is replaced by a constant (i.e., an object in the domain of the variable). Our second rule states that we can safely remove a subset of formulas from the MLN if all equivalence classes of variables in the remaining MLN are single occurrence and all formulas in the subset are tautology (i.e., evaluate to true) at extremes (i.e., assignments with identical truth value for groundings of a predicate). We prove that our two new rules are sound and demonstrate via a detailed experimental evaluation that our approach is superior in terms of scalability and MAP solution quality to the state of the art approaches.

## 1 Introduction

Markov logic [4] uses weighted first order formulas to compactly encode uncertainty in large, relational domains such as those occurring in natural language understanding and computer vision. At a high level, a Markov logic network (MLN) can be seen as a template for generating ground Markov networks. Therefore, a natural way to answer inference queries over MLNs is to construct a ground Markov network and then use standard inference techniques (e.g., Loopy Belief Propagation) for Markov networks. Unfortunately, this approach is not practical because the ground Markov networks can be quite large, having millions of random variables and features.

Lifted inference approaches [17] avoid grounding the whole Markov logic theory by exploiting symmetries in the first-order representation. Existing lifted inference algorithms can be roughly divided into two types: algorithms that lift exact solvers [2, 3, 6, 17], and algorithms that lift approximate inference techniques such as belief propagation [12, 20] and sampling based methods [7, 21]. Another line of work [1, 5, 9, 15] attempts to characterize the complexity of lifted inference independent of the specific solver being used. Despite the presence of large literature on lifting, there has been limited focus on exploiting the specific structure of the MAP problem. Some recent work [14, 16] has looked at exploiting symmetries in the context of LP formulations for MAP inference. Sarkhel et. al [19] show that the MAP problem can be propositionalized in the limited setting of non-shared MLNs. But largely, the question is still open as to whether there can be a greater exploitation of the structure for lifting MAP inference.

In this paper, we propose two new rules for lifted inference specifically tailored for MAP queries. We identify equivalence classes of variables which are single occurrence i.e., they have at most a single variable from the class appearing in any given formula. Our *first rule for lifting* states that MAP inference over the original theory can be equivalently formulated over a reduced theory where every single occurrence class has been reduced to a unary sized domain. This leads to a general framework for transforming the original theory into a (MAP) equivalent reduced theory. Any existing (propositional or lifted) MAP solver can be applied over this reduced theory. When every equivalence class is single occurrence, our approach is domain independent, i.e., the complexity of MAP inference does not depend on the number of constants in the domain. Existing lifting constructs such as the decomposer [6] and the non-shared MLNs [19] are special cases of our single occurrence rule.

When the MLN theory is single occurrence, one of the MAP solutions lies at extreme, namely all groundings of any given predicate have identical values (true/false) in the MAP assignment. Our *second rule for lifting* states that formulas which become tautology (i.e., evaluate to true) at extreme assignments can be ignored for the purpose of MAP inference when the remaining theory is single occurrence. Many difficult to lift formulas such as symmetry and transitivity are easy to handle in our framework because of this rule. Experiments on three benchmark MLNs clearly demonstrate that our approach is more accurate and scalable than the state of the art approaches for MAP inference.

## 2 Background

A first order logic [18] theory is constructed using the `constant`, `variable`, `function` and `predicate` symbols. Predicates are defined over terms as arguments where each term is either a constant, or a variable or a function applied to a term. A formula is constructed by combining predicates using operators such as $\neg$, $\wedge$ and $\vee$. Variables in a first-order theory are often referred to as `Logical Variables`. Variables in a formula can be universally or existentially quantified. A `Knowledge Base` (KB) is a set of formulas. A theory is in Conjunctive Normal Form (CNF) if it is expressed as a conjunction of disjunctive formulas. The process of `(partial) grounding` corresponds to replacing (some) all of the free variables in a predicate or a formula with constants in the theory. In this paper, we assume function-free first order logic theory with Herbrand interpretations [18], and that variables in the theory are implicitly universally quantified.

Markov Logic [4] is defined as a set of pairs $(f_i, w_i)$, where $f_i$ is a formula in first-order logic and $w_i$ is its weight. The weight $w_i$ signifies the strength of the constraint represented by the formula $f_i$. Given a set of constants, an MLN can be seen as a template for constructing ground Markov networks. There is a node in the network for every ground atom and a feature for every ground formula. The probability distribution specified by an MLN is:

$$P(\mathbf{X} = \mathbf{x}) = \frac{1}{Z} \exp \left( \sum_{i:f_i \in F} w_i n_i(\mathbf{x}) \right) \tag{1}$$

where $\mathbf{X} = \mathbf{x}$ specifies an assignment to the ground atoms, the sum in the exponent is taken over the indices of the first order formulas (denoted by $F$) in the theory, $w_i$ is the weight of the $i^{th}$ formula, $n_i(\mathbf{x})$ denotes the number of true groundings of the $i^{th}$ formula under the assignment $\mathbf{x}$, and $Z$ is the normalization constant. A formula $f$ in MLN with weight $w$ can be equivalently replaced by negation of the formula i.e., $\neg f$ with weight $-w$. Hence, without loss of generality, we will assume that all the formulas in our MLN theory have non-negative weights. Also for convenience, we will assume that each formula is either a conjunction or a disjunction of literals.

The MAP inference task is defined as the task of finding an assignment (there could be multiple such assignments) having the maximum probability. Since $Z$ is a constant and $\exp$ is a monotonically increasing function, the MAP problem for MLNs can be written as:

$$\arg \max_{\mathbf{x}} P(\mathbf{X} = \mathbf{x}) = \arg \max_{\mathbf{x}} \sum_{i:f_i \in F} w_i n_i(\mathbf{x}) \tag{2}$$

One of the ways to find the MAP solution in MLNs is to ground the whole theory and then reformulate the problem as a MaxSAT problem [4]. Given a set of weighted clauses (constraints), the goal in MaxSAT is to find an assignment which maximizes the sum of the weights of the satisfied clauses. Any standard solver such as MaxWalkSAT [10] can be used over the ground theory to find the MAP solution. This can be wasteful when there is rich structure present in the network and lifted inference techniques can exploit this structure [11]. In this paper, we assume an MLN theory for the ease of

exposition. But our ideas are easily generalizable to other similar representations such as weighted parfactors [2], probabilistic knowledge bases [6] and WFOMC [5].

## 3 Basic Framework

### 3.1 Motivation

Most existing work on lifted MAP inference adapts the techniques for lifting marginal inference. One of the key ideas used in lifting is to exploit the presence of a *decomposer* [2, 6, 9]. A decomposer splits the theory into identical but independent sub-theories and therefore only one of them needs to be solved. A counting argument can be used when a decomposer is not present [2, 6, 9]. For theories containing upto two logical variables in each clause, there exists a polynomial time lifted inference procedure [5]. Specifically exploiting the structure of MAP inference, Sarkhel et. al [19] show that MAP inference in non-shared MLNs (with no self joins) can be reduced to a propositional problem. Despite all these lifting techniques, there is a larger class of MLN formulas where it is still not clear whether there exists an efficient lifting algorithm for MAP inference. For instance, consider the single rule MLN theory:

$$w_1 \; Parent(X, Y) \wedge Friend(Y, Z) \Rightarrow Knows(X, Z)$$

This rule is hard to lift for any of the existing algorithms since neither the decomposer nor the counting argument is directly applicable. The counting argument can be applied after (partially) grounding $X$ and as a result lifted inference on this theory will be more efficient than ground inference. However, consider adding transitivity to the above theory:

$$w_2 \; Friend(X, Y) \wedge Friend(Y, Z) \Rightarrow Friend(X, Z)$$

This makes the problem even harder because in order to process the new MLN formula via lifted inference, one has to at least ground both the arguments of $Friend$. In this work, we exploit specific properties of MAP inference and develop two new lifting rules, which are able to lift the above theory. In fact, as we will show, MAP inference for MLN containing (exactly) the two formulas given above is *domain independent*, namely, it does not depend on the domain size of the variables.

### 3.2 Notation and Preliminaries

We will use the upper case letters $X, Y, Z$ etc. to denote the variables. We will use the lower case letters $a, b, c$ etc. to denote the constants. Let $\Delta_X$ denote the domain of a variable $X$. We will assume that the variables in the MLN are standardized apart, namely, no two formulas contain the same variable symbol. Further, we will assume that the input MLN is in `normal form` [9]. An MLN is said to be in normal form if a) If $X$ and $Y$ are two variables appearing at the same argument position in a predicate $P$ in the MLN theory, then $\Delta_X = \Delta_Y$. b) There are no constants in any formula. Any given MLN can be converted into the normal form by a series of mechanical operations in time that is polynomial in the size of the MLN theory and the evidence. We will require normal forms for simplicity of exposition. For lack of space, proofs of all the theorems and lemmas marked by (*) are presented in the extended version of the paper (see the supplementary material).

Following Jha et. al [9] and Broeck [5], we define a symmetric and transitive relation over the variables in the theory as follows. $X$ and $Y$ are related if either a) they appear in the same position of a predicate $P$, or b) $\exists$ a variable $Z$ such that $X, Z$ and $Y, Z$ are related. We refer to the relation above as *binding* relation [5]. Being symmetric and transitive, binding relation splits the variables into a set of `equivalence classes`. We say that $X$ and $Y$ bind to each other if they belong to the same equivalence class under the binding relation. We denote this by writing $X \sim Y$. We will use the notation $\bar{X}$ to refer to the equivalence class to which variable $X$ belongs. As an example, the MLN theory consisting of two rules: 1) $P(X) \vee Q(X, Y)$  2) $P(Z) \vee Q(U, V)$ has two variable equivalence classes given by $\{X, Z, U\}$ and $\{Y, V\}$.

Broeck [5] introduce the notion of `domain lifted` inference. An inference procedure is domain lifted if it is polynomial in the size of the variable domains. Note that the notion of domain lifted does not impose any condition on how the complexity depends on the size of the MLN theory. On the similar lines, we introduce the notion of `domain independent` inference.

**Definition 3.1.** *An inference procedure is* `domain independent` *if its time complexity is independent of the domain size of the variables. As in the case of domain lifted inference, the complexity can still depend arbitrarily on the size of the MLN theory.*

# 4 Exploiting Single Occurrence

We show that the domains of equivalence classes satisfying certain desired properties can be reduced to unary sized domains for the MAP inference task. This forms the basis of our first inference rule.

**Definition 4.1.** *Given an MLN theory $M$, a variable equivalence class $\bar{X}$ is said to be* single occurrence *with respect to $M$ if for any two variables $X, Y \in \bar{X}$, $X$ and $Y$ do not appear together in any formula in the MLN. In other words, every formula in the MLN has at most a single occurrence of variables from $\bar{X}$. A predicate is said to be single occurrence if each of the equivalence classes of its argument variables is single occurrence. An MLN is said to be* single occurrence *if each of its variable equivalence classes is single occurrence.*

Consider the MLN theory with two formulas as earlier: 1) $P(X) \vee Q(X, Y)$ 2) $P(Z) \vee Q(U, V)$. Here, $\{Y, V\}$ is a single occurrence equivalence class while $\{X, Z, U\}$ is not. Next, we show that the MAP tuple of an MLN can be recovered from a much smaller MLN in which the domain size of each variable in each single occurrence equivalence class is reduced to one.

## 4.1 First Rule for Lifting MAP

**Theorem 4.1.** *Let $M$ be an MLN theory represented by the set of pairs $\{(f_i, w_i)\}_{i=1}^m$. Let $\bar{X}$ be a single occurrence equivalence class with domain $\Delta_{\bar{X}}$. Then, MAP inference problem in $M$ can be reduced to the MAP inference problem over a simpler MLN $M_{\bar{X}}^r$ represented by a set of pairs $\{(f_i, w_i')\}_{i=1}^m$ where the domain of $\bar{X}$ has been reduced to a single constant.*

*Proof.* We will prove the above theorem by constructing the desired theory $M_{\bar{X}}^r$. Note that $M_{\bar{X}}^r$ has the same set of formulas as $M$ with a set of modified weights. Let $F_{\bar{X}}$ be the set of formulas in $M$ which contain a variable from the equivalence class $\bar{X}$. Let $F_{-\bar{X}}$ be the set of formulas in $M$ which do not contain a variable from the equivalence class $\bar{X}$. Let $\{a_1, a_2, \ldots, a_r\}$ be the domain of $\bar{X}$. We will split the theory $M$ into $r$ equivalent theories $\{M_1, M_2, \ldots, M_r\}$ such that for each $M_j$: [1]

**1.** For every formula $f_i \in F_{\bar{X}}$ with weight $w_i$, $M_j$ contains $f_i$ with weight $w_i$.
**2.** For every formula $f_i \in F_{-\bar{X}}$ with weight $w_i$, $M_j$ contains $f_i$ with weight $w_i/r$.
**3.** Domain of $\bar{X}$ in $M_j$ is reduced to a single constant $\{a_j\}$.
**4.** All other equivalence classes have domains identical to that in $M$.
This divides the set of weighted constraints in $M$ across the $r$ sub-theories. Formally:

**Lemma 4.1.*** *The set of weighted constraints in $M$ is a union of the set of weighted constraints in the sub-theories $\{M_j\}_{j=1}^r$.*

**Corollary 4.1.** *Let $\mathbf{x}$ be an assignment to the ground atoms in $M$. Let the function $W_M(\mathbf{x})$ denote the weight of satisfied ground formulas in $M$ under the assignment $\mathbf{x}$ in theory $M$. Further, let $\mathbf{x_j}$ denote the assignment $\mathbf{x}$ restricted to the ground atoms in theory $M_j$. Then: $W_M(\mathbf{x}) = \sum_{j=1}^r W_{M_j}(\mathbf{x_j})$.*

It is easy to see that $M_j$'s are identical to each other upto the renaming of the constants $a_j$'s. Hence, exploiting symmetry, there is a one to one correspondence between the assignments across the sub-theories. In particular, there is one to one correspondence between MAP assignments across the sub-theories $\{M_j\}_{j=1}^r$.

**Lemma 4.2.** *If $\mathbf{x_j^{MAP}}$ is a MAP assignment to the theory $M_j$, then there exists a MAP assignment $\mathbf{x_l^{MAP}}$ to $M_l$ such that $\mathbf{x_l^{MAP}}$ is identical to $\mathbf{x_j^{MAP}}$ with the difference that occurrence of constant $a_j$ (in ground atoms of $M_j$) is replaced by constant $a_l$ (in ground atoms of $M_l$).*

Proof of this lemma follows from the construction of the sub-theories $M_1, M_2, \ldots M_r$. Next, we will show that MAP solution for the theory $M$ can be read off from the MAP solution for any of theories $\{M_j\}_{j=1}^r$. Without loss of generality, let us consider the theory $M_1$. Let $\mathbf{x_1^{MAP}}$ be some MAP assignment for $M_1$. Using lemma 4.2 there are MAP assignments $\mathbf{x_2^{MAP}}, \mathbf{x_3^{MAP}}, \ldots, \mathbf{x_r^{MAP}}$ for $M_2, M_3, \ldots M_r$ which are identical to $\mathbf{x_1^{MAP}}$ upto renaming of the constant $a_1$. We construct an assignment $\mathbf{x^{MAP}}$ for the original theory $M$ as follows.

**1.** For each predicate $P$ which does not contain any occurrence of the variables from the equivalence class $\bar{X}$, read off the assignment to its groundings in $\mathbf{x^{MAP}}$ from $\mathbf{x_1^{MAP}}$. Note that assignments of groundings of $P$ are identical in each of $x_j^{MAP}$ because of Lemma 4.2.

**2.** The (partial) groundings of each predicate $P$ whose arguments contain a variable $X \in \bar{X}$ are split across the sub-theories $\{M_j\}_{1 \leq j \leq r}$ corresponding to the substitutions $\{X = a_j\}_{1 \leq j \leq r}$, respectively. We assign the groundings of $P$ in $\mathbf{x^{MAP}}$ the values from the assignments $\mathbf{x_1^{MAP}}, \mathbf{x_2^{MAP}}, \ldots \mathbf{x_r^{MAP}}$ for the respective partial groundings. Because of Lemma 4.2, these partial groundings have identical values across the sub-theories upto renaming of the constant $a_j$ and hence, can be read off from either of the sub-theory assignments, and more specifically, $\mathbf{x_1^{MAP}}$.

By construction, assignment $\mathbf{x^{MAP}}$ restricted to the ground atoms in sub-theory $M_j$ corresponds to the assignment $\mathbf{x_j^{MAP}}$ for each $j$, $1 \leq j \leq r$.

The only thing remaining to show is that $\mathbf{x^{MAP}}$ is indeed a MAP assignment for $M$. Suppose it is not, then there is another assignment $\mathbf{x^{alt}}$ such that $W_M(\mathbf{x^{alt}}) > W_M(\mathbf{x^{MAP}})$. Using Corollary 4.1, $W_M(\mathbf{x^{alt}}) > W_M(\mathbf{x^{MAP}}) \Rightarrow \sum_{j=1}^r W_{M_j}(\mathbf{x_j^{alt}}) > \sum_{j=1}^r W_{M_j}(\mathbf{x_j^{MAP}})$. This means that $\exists j$, such that $W_{M_j}(\mathbf{x_j^{alt}}) > W_{M_j}(\mathbf{x_j^{MAP}})$. But this would imply that $\mathbf{x_j^{MAP}}$ is not a MAP assignment for $M_j$ which is a contradiction. Hence, $\mathbf{x^{MAP}}$ is indeed a MAP assignment for $M$. $\square$

**Definition 4.2.** *Application of Theorem 4.1 to transform the MAP problem over an MLN theory $M$ into the MAP over a reduced theory $M_{\bar{X}}^r$ is referred to as Single Occurrence Rule for lifted MAP.*

Decomposer [6] is a very powerful construct for lifted inference. The next theorem states that a decomposer is a single occurrence equivalence class (and therefore, the single occurrence rule includes the decomposer rule as a special case).

**Theorem 4.2.** * *Let $M$ be an MLN theory and let $\bar{X}$ be an equivalence class of variables. If $\bar{X}$ is a decomposer for $M$, then $\bar{X}$ is single occurrence in $M$.*

### 4.2 Domain Independent Lifted MAP

A simple procedure for lifted MAP inference which utilizes the property of MLN reduction for single occurrence equivalence classes is given in Algorithm 1. Here, the MLN theory is successively reduced with respect to each of the single occurrence equivalence classes.

---

**Algorithm 1** Reducing all the single occurrence equivalence classes in an MLN

---

**reduce(MLN $M$)**
  $M^r \leftarrow M$
  **for all** Equivalence-Class $\bar{X}$ **do**
    **if** (isSingleOccurrence($\bar{X}$)) **then**
      $M^r \leftarrow$ reduceEQ($M^r,\bar{X}$)
    **end if**
  **end for**
  return $M^r$

**reduceEQ(MLN M, class $\bar{X}$)**
  $M_{\bar{X}}^r \leftarrow \{\}$; $size \leftarrow |\Delta_{\bar{X}}|$; $\Delta_{\bar{X}} \leftarrow \{a_1^{\bar{X}}\}$
  **for all** Formulas $f_i \in F_{\bar{X}}$ **do**
    Add $(f_i, w_i)$ to $M_{\bar{X}}^r$
  **end for**
  **for all** Formulas $f_i \in F_{-\bar{X}}$ **do**
    Add $(f_i, w_i/size)$ to $M_{\bar{X}}^r$
  **end for**;   return $M_{\bar{X}}^r$

---

**Theorem 4.3.** * *MAP inference in a single occurrence MLN is domain independent.*

If an MLN theory contains a combination of both single occurrence and non-single occurrence equivalence classes, we can first reduce all the single occurrence classes to unary domains using Algorithm 1. Any existing (lifted or propositional) solver can be applied on this reduced theory to obtain the MAP solution. Revisiting the single rule example from Section 3.1: $Parent(X,Y) \wedge Friend(Y,Z) \Rightarrow Knows(X,Z)$, we have 3 equivalence classes $\{X\}$, $\{Y\}$, and $\{Z\}$, all of which are single occurrence. Hence, MAP inference for this MLN theory is domain independent.

## 5 Exploiting Extremes

Even when a theory does not contain single occurrence variables, we can reduce it effectively if a) there is a subset of formulas all of whose groundings are satisfied at extremes i.e. the assignments with identical truth value for all the groundings of a predicate, and b) the remaining theory with these formulas removed is single occurrence. This is the key idea behind our second rule for lifted MAP. We will first formalize the notion of an extreme assignment followed by the description of our second lifting rule.

## 5.1 Extreme Assignments

**Definition 5.1.** *Let $M$ be an MLN theory. Given an assignment $\mathbf{x}$ to the ground atoms in $M$, we say that predicate $P$ is at extreme in $\mathbf{x}$ if all the groundings of $P$ take the same value (either* `true` *or* `false`*) in $\mathbf{x}$. We say that $\mathbf{x}$ is at extreme if all the predicates in $M$ are at extreme in $\mathbf{x}$.*

**Theorem 5.1.\*** *Given an MLN theory $M$, let $\mathbf{P_S}$ be the set of predicates which are single occurrence in $M$. Then there is a MAP assignment $\mathbf{x^{MAP}}$ such that $\forall P \in \mathbf{P_S}$, $P$ is at extreme in $\mathbf{x^{MAP}}$.*

**Corollary 5.1.** *A single occurrence MLN admits a MAP solution which is at extreme.*

Sarkhel et. al [19] show that non-shared MLNs (with no self-joins) have a MAP solution at the extreme. This turns out to be a special case of single occurrence MLNs.

**Theorem 5.2.\*** *If an MLN theory is non-shared and has no-self joins, then $M$ is single occurrence.*

## 5.2 Second Rule for Lifting MAP

Consider the MLN theory with a single formula as in Section 3.1: $w_1$ $Parent(X, Y) \wedge Friend(Y, Z) \Rightarrow Knows(X, Z)$. This is a single occurrence MLN and hence by Corollary 5.1, MAP solution lies at extreme. Consider adding the transitivity constraint to the theory: $w_2$ $Friend(X, Y) \wedge Friend(Y, Z) \Rightarrow Friend(X, Z)$. All the groundings of the second formula are satisfied at any extreme assignment of the $Friends$ predicate groundings. Since, the MAP solution to the original theory with single formula is at extreme, it satisfies all the groundings of the second formula. Hence, it is a MAP for the new theory as well. We introduce the notion of tautology at extremes:

**Definition 5.2.** *An MLN formula $f$ is said to be a tautology at extremes if all of its groundings are satisfied at any of the extreme assignments of its predicates.*

If an MLN theory becomes single occurrence after removing all the tautologies at extremes in it, then MAP inference in such a theory is domain independent.

**Theorem 5.3.\*** *Let $M$ be an MLN theory with the set of formulas denoted by $F$. Let $F_{te}$ denote a set of formulas in $M$ which are tautologies at extremes. Let $M'$ be a new theory with formulas $F - F_{te}$ and formula weights as in $M$. Let the variable domains in $M'$ be same as in $M$. If $M'$ is single occurrence then the MAP inference for the original theory $M$ can be reduced to the MAP inference problem over the new theory $M'$.*

**Corollary 5.2.** *Let $M$ be an MLN theory. Let $M'$ be a single occurrence theory (with variable domains identical to $M$) obtained after removing a subset of formulas in $M$ which are tautologies at extremes. Then, MAP inference in $M$ is domain independent.*

**Definition 5.3.** *Application of Theorem 5.3 to transform the MAP problem over an MLN theory $M$ into the MAP problem over the remaining theory $M'$ after removing (a subset of) tautologies at extremes is referred to as Tautology at Extremes Rule for lifted MAP.*

Clearly, Corollary 5.2 applies to the two rule MLN theory considered above (and in the Section 3.1) and hence, MAP inference for the theory is domain independent. A necessary and sufficient condition for a clausal formula to be a tautology at extremes is to have both positive and negative occurrences of the same predicate symbol. Many difficult to lift but important formulas such as symmetry and transitivity are tautologies at extremes and hence, can be handled by our approach.

## 5.3 A Procedure for Identifying Tautologies

In general, we only need the equivalence classes of variables appearing in $F_{te}$ to be single occurrence in the remaining theory for Theorem 5.3 to hold. [2] Algorithm 2 describes a procedure to identify the largest set of tautologies at extremes such that all the variables in them are single occurrence with respect to the remainder of the theory. The algorithm first identifies all the tautologies at extremes. It then successively removes those from the set all of whose variables are not single occurrence in the remainder of the theory. The process stops when all the tautologies have only single occurrence variables appearing in them. We can then apply the procedure in Section 4 to find the MAP solution for the remainder of the theory. This is also the MAP for the whole theory by Theorem 5.3.

**Algorithm 2** Finding Tautologies at Extremes with Single Occurrence Variables

---

**getSingleOccurTautology(MLN $M$)**
  $F_{te} \leftarrow$ getAllTautologyAtExtremes($M$);
  $F' = F - F_{te}$; fixpoint=False;
  **while** (fixpoint==False) **do**
    EQVars $\leftarrow$ getSingleOccurVars($F'$)
    fixpoint=True
    **for all** formulas $f \in F_{te}$ **do**
      **if** (!(Vars(f) $\subseteq$ EQVars)) **then**
        $F' \leftarrow F' \cup \{f\}$; fixpoint = False
      **end if**
    **end for**
  **end while**;    return $F - F'$

**getAllTautologyAtExtremes(MLN $M$)**
  //Iterate over all the formulas in $M$ and return the
  //subset of formulas which are tautologies at extremes
  *//Pseudocode omitted due to lack of space*

**isTautologyAtExtreme(Formula $f$)**
  $f' =$ Clone($f$)
  $\mathbf{P}_U \leftarrow$ set of unique predicates in $f'$
  **for all** $P \in \mathbf{P}_U$ **do**
    ReplaceByNewPropositionalPred($P$,$f'$)
  **end for**
  // $f'$ is a propositional formula at this point
  return isTautology($f'$)

---

# 6 Experiments

We compared the performance of our algorithm against Sarkhel et. al [19]'s non shared MLN approach and the purely grounded version on three benchmark MLNs. For both the lifted approaches, we used them as pre-processing algorithms to reduce the MLN domains. We applied the ILP based solver Gurobi [8] as the base solver on the reduced theory to find the MAP assignment. In principle, any MAP solver could be used as the base solver [3]. For the ground version, we directly applied Gurobi on the grounded theory. We will refer to the grounded version as GRB. We will refer to our and Sarkhel et. al [19]'s approaches as SOLGRB (*Single Occurrence Lifted GRB*) and NSLGRB (*Non-shared Lifted GRB*), respectively.

## 6.1 Datasets and Methodology

We used the following benchmark MLNs for our experiments. (Results on the Student network [19] are presented in the supplement.):
1) **Information Extraction (IE):** This theory is available from the Alchemy [13] website. We pre-processed the theory using the pure literal elimination rule described by Sarkhel et. al [19]. Resulting MLN had 7 formulas, 5 predicates and 4 variable equivalence classes.
2) **Friends & Smokers (FS):** This is a standard MLN used earlier in the literature [20]. The MLN has 2 formulas, 3 predicates and 1 variable equivalence class. We also introduced singletons for each predicate.

For each algorithm, we report:
1) **Time:** Time to reach the optimal as the domain size is varied from 25 to 1000. [4,5]
2) **Cost:** Cost of the unsatisfied clauses as the running time is varied for a fixed domain size (500).
3) **Theory Size:** Ground theory size as the domain size is varied.

All the experiments were run on an Intel four core i3 processor with 4 GB of RAM.

## 6.2 Results

Figures 1a-1c plot the results for the IE domain. Figure 1a shows the time taken to reach the optimal. [6] This theory has a mix of single occurrence and non-single occurrence variables. Hence, every algorithm needs to ground some or all of the variables. SOLGRB only grounds the variables whose domain size was kept constant. Hence, varying domain size has no effect on SOLGRB and it reaches optimal instantaneously for all the domain sizes. NSLGRB partially grounds this theory and its time to optimal gradually increases with increasing domain size. GRB performs significantly worse due to grounding of the whole theory.

Figure 1b (log scale) plots the total cost of unsatisfied formulae with varying time at domain size of 500. SOLGRB reaches optimal right in the beginning because of a very small ground theory. NSLGRB takes about 15 seconds. GRB runs out of memory. Figure 1c (log scale) shows the size of the ground theory with varying domain size. As expected, SOLGRB stays constant whereas the

ground theory size increases polynomially for both NSLGRB and GRB with differing degrees (due to the different number of variables grounded).

Figure 2 shows the results for FS. This theory is not single occurrence but the tautology at extremes rule applies and our theory does not need to ground any variable. NSLGRB is identical to the grounded version in this case. Results are qualitatively similar to IE domain. Time taken to reach the optimal is much higher in FS for NSLGRB and GRB for larger domain sizes.

These results clearly demonstrate the scalability as well as the superior performance of our approach compared to the existing algorithms.

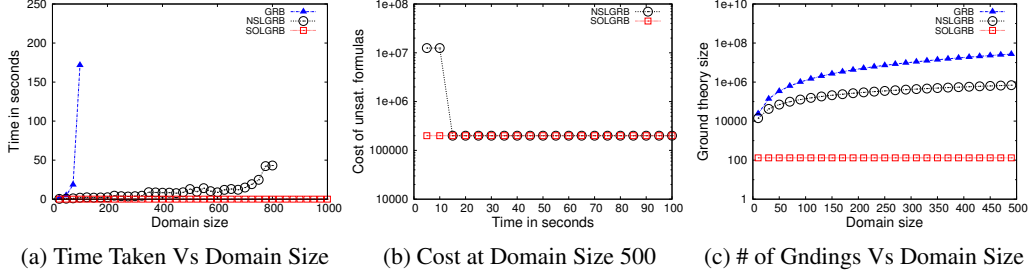

| (a) Time Taken Vs Domain Size | (b) Cost at Domain Size 500 | (c) # of Gndings Vs Domain Size |

Figure 1: IE

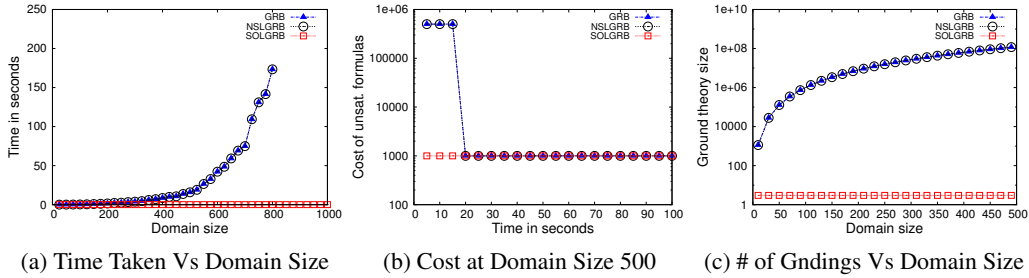

| (a) Time Taken Vs Domain Size | (b) Cost at Domain Size 500 | (c) # of Gndings Vs Domain Size |

Figure 2: Friends & Smokers

# 7 Conclusion and Future Work

We have presented two new rules for lifting MAP inference which are applicable to a wide variety of MLN theories and result in highly scalable solutions. The MAP inference problem becomes domain independent when every equivalence class is single occurrence. In the current framework, our rules have been used as a pre-processing step generating a reduced theory over which any existing MAP solver can be applied. This leaves open the question of effectively combining our rules with existing lifting rules in the literature.

Consider the theory with two rules: $S(X) \lor R(X)$ and $S(Y) \lor R(Z) \lor T(U)$. Here, the equivalence class $\{X, Y, Z\}$ is not single occurrence, and our algorithm will only be able to reduce the domain of equivalence class $\{U\}$. But if we apply Binomial rule [9] on $S$, we get a new theory where $\{X, Z\}$ becomes a single occurrence equivalence class and we can resort to domain independent inference. [7] Therefore, application of Binomial rule before single occurrence would lead to larger savings. In general, there could be arbitrary orderings for applying lifted inference rules leading to different inference complexities. Exploring the properties of these orderings and coming up with an optimal one (or heuristics for the same) is a direction for future work.

# 8 Acknowledgements

Happy Mittal was supported by TCS Research Scholar Program. Vibhav Gogate was partially supported by the DARPA Probabilistic Programming for Advanced Machine Learning Program under AFRL prime contract number FA8750-14-C-0005. We are grateful to Somdeb Sarkhel and Deepak Venugopal for sharing their code and also for helpful discussions.

## Footnotes

[1]Supplement presents an example of splitting an MLN theory based on the following procedure.

[2]Theorem $5.3^g$ in the supplement gives a more general version of Theorem 5.3.

[3]Using MaxWalkSAT [10] as the base solver resulted in sub-optimal results.

[4]For IE, two of the variable domains of were varied and other two were kept constant at 10 as done in [19].

[5]Reported results are averaged over 5 runs.

[6] NSLGRB and GRB ran out of memory at domain sizes 800 and 100, respectively.

[7] A decomposer does not apply even after conditioning on $S$.

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
