[Supplementary Material]

# Supplementary Material: New Rules for Domain Independent Lifted MAP Inference

**Happy Mittal, Prasoon Goyal**
Dept. of Comp. Sci. & Engg.
I.I.T. Delhi, Hauz Khas
New Delhi, 110016, India
happy.mittal@cse.iitd.ac.in
prasoongoyal13@gmail.com

**Vibhav Gogate**
Dept. of Comp. Sci.
Univ. of Texas Dallas
Richardson, TX 75080, USA
vgogate@hlt.utdallas.edu

**Parag Singla**
Dept. of Comp. Sci. & Engg.
I.I.T. Delhi, Hauz Khas
New Delhi, 110016, India
parags@cse.iitd.ac.in

## 4 Exploiting Single Occurrence

### 4.1 Splitting the Theory: Example

Consider the MLN theory $M$ consisting of three formulas:
$w_1 : P(X) \vee Q(X, Y)$
$w_2 : P(Z) \vee Q(U, V)$
$w_3 : R(W)$

Here the first two formulas are as considered earlier in the main text. We have also added another formula $R(W)$ to the theory. Let the domain of each of the variables be $\Delta = \{a, b, c\}$. This theory consists of 3 equivalence classes given by $E_1 = \{X, Z, U\}$ and $E_2 = \{Y, V\}$ and $E_3 = \{W\}$. Here, $E_2$ and $E_3$ are single occurrence classes whereas $E_1$ is not.

Let us consider splitting the theory based on the groundings of the equivalence class $E_2$. This results in three sub-theories $M_1$, $M_2$ and $M_3$ with the domain of $E_2$ being split as $\Delta_{E_2} = \{a\}, \{b\}$, and $\{c\}$ in $M_1$, $M_2$ and $M_3$, respectively. Domains of $E_1$ and $E_3$ remain the same as in $M$. Figure 1 depicts each of the sub-theories along with the formula weights. First two formulas contain an occurrence of a variable from the class $E_2$ and hence, their weights in the sub-theories remain the same as in $M$. Last formula does not have contain variable from the equivalence class $E_2$, hence, the corresponding weight gets divided by $|\Delta_{E_2}| = 3$ in each of the sub-theories.

It is easy to see that the set of weighted constraints in $M$ is a union of the set of the weighted constraints in the sub-theories $M_1$, $M_2$ and $M_3$. Consider the groundings of the first two formulas which contain a variable from the set $E_2$. Each grounding of these two formulas is present in exactly one of the sub-theories with the same weight as in the original theory. Hence, the claim holds for the first two formulas. Next, consider the last formula which does not contain any variable from the set $E_2$. In this case, all the formula groundings are replicated in all the 3 sub-theories with weight $w_3/3$, $w_3$ being the weight in the original theory. So, the total weight for each of the groundings in the union remains the same. Hence, proved.

| $w_1 : P(X) \vee Q(X, Y)$ <br> $w_2 : P(Z) \vee Q(U, V)$ <br> $\frac{w_3}{3} : R(W)$ | $w_1 : P(X) \vee Q(X, Y)$ <br> $w_2 : P(Z) \vee Q(U, V)$ <br> $\frac{w_3}{3} : R(W)$ | $w_1 : P(X) \vee Q(X, Y)$ <br> $w_2 : P(Z) \vee Q(U, V)$ <br> $\frac{w_3}{3} : R(W)$ |
|---|---|---|
| (a) $M_1$ with $\Delta_{E_2} = \{a\}$ | (b) $M_2$ with $\Delta_{E_2} = \{b\}$ | (c) $M_3$ with $\Delta_{E_2} = \{c\}$ |

Figure 1: Splitting an MLN theory

### 4.2 Proofs Omitted from the Main Text

**Lemma 4.1.** *The set of weighted constraints in $M$ is a union of the set of weighted constraints in the sub-theories $\{M_j\}_{j=1}^r$.*

*Proof.* For each $f_i \in F_{\bar{X}}$, $f_i$ has exactly one occurrence of a variable from the equivalence class $\bar{X}$. Without loss of generality, let the variable $X \in \bar{X}$ which appears in $f_i$ be $X$. Then, $f_i$ has exactly one (partial) grounding corresponding to each of the constants in the domain of $X$ i.e. $\{a_1, a_2, \dots a_r\}$. These groundings are split across the sub-theories $M_1, M_2 \dots M_r$ with $M_j$ containing the (partial) grounding corresponding to the constant $a_j$. Weight of each of these (partial) groundings is $w_i$ as in the original MLN. For each formula $f_i \in F_{-\bar{X}}$ with weight $w_i$, $f_i$ is present in each sub-theory with weight $w_i/r$. Since there are $r$ such theories, the total weight of $f_i$ in the sub-theories adds upto $w_i$ [1]. Hence, the set of (weighted) constraints is a union of the set of constraints in the sub-theories $M_1, M_2, \dots M_r$. $\square$

**Theorem 4.2.** *Let $M$ be an MLN theory and let $\bar{X}$ be an equivalence class of variables. If $\bar{X}$ is a decomposer for $M$, then $\bar{X}$ is single occurrence in $M$.*

*Proof.* Let $F$ be the set of formulas in $M$. Following Gogate & Domingos [1] (Definition 2), for $\bar{X}$ to be a decomposer we require: 1) If $X \in \bar{X}$ appears in a formula $f \in F$, then $X$ appears in every predicate in $f$ 2) If $X, Y \in \bar{X}$ then they can not appear at different positions of any predicate $P$. Clearly, for any two variables $X, Y \in \bar{X}$, they can not appear together in the same formula. Because if they did, they will have to appear in every predicate of the formula and at the same position. This is not possible. Hence, $\bar{X}$ must be single occurrence in $M$. Additionally, it follows that a decomposer can alternately be defined as a variable equivalence class $\bar{X}$ such that 1) $\bar{X}$ is single occurrence 2) If $X \in \bar{X}$ appears in a formula $f \in F$, then $X$ appears in every predicate in $f$. $\square$

**Theorem 4.3.** *MAP inference in a single occurrence MLN is domain independent.*

*Proof.* Since MLN is single occurrence, every variable equivalence class in the MLN is single occurrence. Using Algorithm 1, we can successively reduce the MAP problem over original MLN to another MLN $M^r$ where domain of every variable has been reduced to unary. Running MAP over $M^r$ gives the MAP for the original problem.

There is one technicality here regarding how the MAP solution is represented. If we represent the MAP value for each ground atom explicitly, then generating the output itself will take time polynomial in the domain sizes of the variables [2]. Therefore, MAP solution will have to be represented in a first-order form where the solution is specified for each first order predicate rather than each ground atom. But this is straightforward to do in the above setting. $\square$

## 5 Exploiting Extremes

### 5.1 Proofs Omitted from the Main Text

**Theorem 5.1.** *Given an MLN theory $M$, let $\mathbf{P}_S$ be the set of predicates which are single occurrence in $M$. Then there is a MAP assignment $\mathbf{x^{MAP}}$ such that $\forall P \in \mathbf{P}_S$, $P$ is at extreme.*

*Proof.* Reduce the MLN theory $M$ using the Algorithm 1 given in main text. Let $M^r$ be the resulting theory. Since every $P \in P_S$ contains only single occurrence binding classes, domain of each argument of $P$ is reduced to a singleton in $M^r$. Since MAP solution for $M$ can be read off from the MAP solution for $M^r$, each grounding of $P$ will get identical values in the MAP solution for $M$. $\square$

**Theorem 5.2.** *If an MLN theory is non-shared and has no-self joins, then $M$ is single occurrence.*

*Proof.* Since MLN is non shared, every equivalence class consists of variables appearing only in a single predicate at a particular position. Further, since there are no self joins, a formula can not have two variables $X, Y$ from the same equivalence class. Because if it did, they will have to appear either in different positions of the same predicate or in different predicates which is a contradiction. $\quad\square$

**Theorem 5.3.** *Let $M$ be an MLN theory with the set of formulas denoted by $F$. Let $F_{te}$ denote a set of formulas in $M$ which are tautologies at extremes. Let $M'$ be a new theory with formulas $F - F_{te}$ and formula weights as in $M$. Let the variable domains in $M'$ be same as in $M$. If $M'$ is single occurrence then the MAP inference for the original theory $M$ can be reduced to the MAP inference problem over the new theory $M'$.*

*Proof.* Let $M_{te}$ denote an MLN theory consisting of the formulas in the set $F_{te}$ with weights as in $M$. Clearly, $M$ is a union of the weighted constraints in the theories $M'$ and $M_{te}$.

Let us first assume that all the predicates appearing in the theory $M$ also appear in $M'$. We will relax this assumption later. Since $M'$ is single occurrence, there is MAP solution for $M'$ which is at extreme by Corollary 5.1. Let $\mathbf{x'^{MAP}}$ be such a MAP assignment. Since all the predicates in $M$ also appear in $M'$, and the variable domains are the same in both $M$ and $M'$, $\mathbf{x'^{MAP}}$ is also a valid assignment for $M$. For ease of notation, we will interchangeably refer to this assignment as $\mathbf{x^{MAP}}$. By definition, $\mathbf{x'^{MAP}}$ is the restriction of $\mathbf{x^{MAP}}$ to the ground atoms in $M'$. Let $\mathbf{x_{te}^{MAP}}$ denote the restriction of $\mathbf{x^{MAP}}$ to the ground atoms in $M_{te}$.

We will show that $\mathbf{x^{MAP}}$ is a MAP assignment for $M$. Let there be another MAP assignment $\mathbf{x^{alt}}$ for $M$ such that $W_M(\mathbf{x^{alt}}) > W_M(\mathbf{x^{MAP}})$. Let $\mathbf{x'^{alt}}$ and $\mathbf{x_{te}^{alt}}$ be the restrictions of $\mathbf{x^{alt}}$ to the ground atoms in $M'$ and $M_{te}$, respectively. Since $M$ is a union of the weighted constraints in $M'$ and $M_{te}$, $W_M(\mathbf{x^{alt}}) > W_M(\mathbf{x^{MAP}}) \Rightarrow \left( W_{M'}(\mathbf{x'^{alt}}) + W_{M_{te}}(\mathbf{x_{te}^{alt}}) \right) > \left( W_{M'}(\mathbf{x'^{MAP}}) + W_{M_{te}}(\mathbf{x_{te}^{MAP}}) \right)$. But $W_{M_{te}}(\mathbf{x_{te}^{alt}}) \leq W_{M_{te}}(\mathbf{x_{te}^{MAP}})$ since $\mathbf{x_{te}^{MAP}}$ is an extreme assignment and all the groundings of formulas in $M_{te}$ are satisfied at extremes (and hence, any other assignment can not be any better than $\mathbf{x_{te}^{MAP}}$ for $M_{te}$). Therefore, we should have $W_{M'}(\mathbf{x'^{alt}}) > W_{M'}(\mathbf{x'^{MAP}})$ which is a contradiction since $\mathbf{x'^{MAP}}$ was a MAP assignment for $M'$. Hence, $\mathbf{x^{MAP}}$ must be a MAP assignment for $M$.

Next, consider the case when predicates in $M'$ are a strict subset of those in $M$. Let $P^{'-}$ be the subset of predicates which are in $M$ but not in $M'$. Then, we can extend $\mathbf{x'^{MAP}}$ to $\mathbf{x^{MAP}}$ by assigning any extreme assignment to the groundings of $P^{'-}$ and keeping rest of the assignment same as in $\mathbf{x'^{MAP}}$. Rest of the proof above follows as before. $\quad\square$

**Theorem 5.3.$^{\text{g}}$** *Let $M$ be an MLN theory with the set of formulas denoted by $F$. Let $F_{te}$ denote a set of formulas in $M$ which are tautologies at extremes. Let $M'$ be a new theory with formulas $F - F_{te}$ and formula weights as in $M$. Let the variable domains in $M'$ be same as in $M$. Let $\mathbf{\bar{X}_{te}}$ denote the set of equivalence classes for the variables appearing in the formulas $F_{te}$. If $\forall \bar{X} \in \mathbf{\bar{X}_{te}}$, $\bar{X}$ is single occurrence in $M'$ then the MAP inference for the original theory $M$ can be reduced to the MAP inference problem over the new theory $M'$.*

*Proof.* As in Theorem 5.3, let $M_{te}$ denote an MLN theory consisting of the formulas in the set $F_{te}$ with weights as in $M$. Let $P_{te}$ denote the set of predicates appearing the theory $M_{te}$. As earlier, $M$ is a union of the weighted constraints in the theories $M'$ and $M_{te}$.

In this general version of the theorem, $M'$ may not be single occurrence any more. So, the MAP solution for $M'$ may not be at extreme in general. But note that we only care about predicates in the set of formulas $F_{te}$ to be at extreme, for the tautologies to hold.

Now, consider any predicate $P \in P_{te}$. Recall that $P$ only contains variables from the set $\mathbf{\bar{X}_{te}}$ where every equivalence class in the set $\mathbf{\bar{X}_{te}}$ is single occurrence in $M'$. In other words, $P$ is single occurrence in $M'$. Hence, by Theorem 5.1, there exists a MAP assignment $\mathbf{x'^{MAP}}$ for $M'$ such that $P$ is at extreme in $\mathbf{x'^{MAP}}$. [3] Hence, all the groundings of the formulas in the set $F_{te}$ will be satisfied in $\mathbf{x'^{MAP}}$. We can extend $\mathbf{x'^{MAP}}$ to $\mathbf{x^{MAP}}$ by assigning the same values as in $\mathbf{x'^{MAP}}$ for predicates which appear in $M'$ and assigning any extreme value to the remaining predicates.

Hence, $\mathbf{x}^{\mathbf{MAP}}$ is an assignment to $M$ such that its restriction to $M'$ and $M_{te}$ results in the MAP assignment for the respective theories. Rest of the proof follows from the proof of Theorem 5.3

$\square$

# 6  Experiments

**Student Network (Student):** We used exactly the same MLN (and weights) as done by Sarkhel et. al [2] in their experiments. The MLN theory has $4$ formulas, $3$ predicates and $4$ variable equivalence classes. This is a single occurrence MLN and SOLGRB does need to ground any of the variables. NSLGRB results in partial grounding of this domain.

Figure 2a shows the time taken to reach the optimal as the domain size is varied. As expected, SOLGRB reaches the optimal in practically no time independent of the domain size. NSLGRB does not give optimal result beyond the domain size of $125$. [4] GRB runs out of memory beyond domain size $25$. [5]

Figure 2b plots the cost of unsatisfied formulas for domain size $500$ as the run time is varied. SOL-GRB reaches the optimal right at the beginning of the run. NSLGRB always results in highly sub-optimal solutions. GRB does not scale to this domain size. Figure 2c shows the results for the size of the ground theory with varying domain size. This is similar to the one obtained for IE.

(a) Time Taken Vs Domain Size    (b) Cost at Domain Size 500    (c) # of Gndings Vs Domain Size

Figure 2: student

## Footnotes

[1] Since $f_i$ does not contain any variable from the equivalence class $\bar{X}$, the domain of its variables is unaffected in the sub-theories

[2] The degree of the polynomial depends on the maximum number of variables in a formula.

[3] If $P$ does not belong to the theory $M'$, we can consider any extreme assignment for $P$.

[4]The graph plots the time usage only for the domain sizes for which optimal was obtained.

[5]There is a single point for GRB in the graph.