[Reviews · NeurIPS 2014]

Submitted by Assigned_Reviewer_17

This paper introduces two new lifted inference rules for MAP. They apply when there are -for the MPA task- redundant logical variables, and when formulas are always true when entire relations are set to true/false.

The definition of domain independence should be made more precise. In its current form, there is a problem. If the MAP task is to compute a probability, the complexity cannot be independent of the domain, because the probability is a function of it. If the MAP problem is to compute a MAP state, then a database representation of that state also has size dependent on the domain. This problem is fixed if you allow your output to be a first-order formula. For example, your output could be FORALL x y Friends(x,y).

The experiments suggest that the proposed rules apply to non-trivial MLNs. This is a weakness of the paper though: can you apply these rules to a large, real MLN? Do they still apply as the MLN gets more complex, with more formulas. This is not clear at the moment. In particular, I worry about formulas of length 1. These are always in the MLN, often with negative weights, and does that not break the required properties? For example, if you add to the transitivity model the formula
1 !Friends(x,y)
What happens then?

I believe there is a connection between extreme assignments and the formulas being "montone" or having "safe negation".

It is strange to talk about functions in logic and then not use them.

It should be noted that the replacement of existentially quantified variables on line 79 breaks domain-independence.

Line 131 should say "containing the two formulas _and no others_".

Line 312: drop "necessary"?

Summary: The paper aims to be simple and intuitive, which I appreciate. The presentation is good. The proposed rules are quite simple, yet they apply to interesting unsolved cases. Overall I like the paper.

Submitted by Assigned_Reviewer_32

This paper introduces two new rules for lifted inference to speed up MAP queries in Markov Logic Networks. The first rule uses domain reduction (to size 1) for variables with single occurrences in formulas. The second rule removes formulas that have become tautologies under extreme assignments. The framework generalizes and significantly extends previous approaches and can handle key constructs such as symmetries and transitive (in contrast to previous work).

I did not check all technical details but approach appears sound.

Experiments show significantly better scaling compared to previous methods.
Summary: Paper introduces two new rules for lifted inference for MAP queries in Markove Logic Networks. Rules generalize and extend previous approaches. Experiments show much better scaling properties.

Submitted by Assigned_Reviewer_41

Summary:
This paper proposes two rules for lifted inference in MLNs by exploiting the structure for MAP queries.
The first rule is for applying MAP inference over a reduced theory such that single occurrences in equivalence classes are reduced to a unary sized domain.
The second rule applies to the single occurrence theory, ignoring the tautology formulas for extreme assignments.

Quality:
This paper has a good quality. All theorems and lemmas have proofs in the paper or in the supported document.
While the experimental results seem very promising, they are confusing with respect to the NSLGRB approach (Sarkhel et. al paper). In their paper, the results show constant growth w.r.t the domain size and time, however the pictures here show that the cost increases in time and domain size. Is there any justification of how the results presented in this paper are different from the claim of the original paper? Maybe the number of domain sizes (100 in their claims vs 500)? I am not convinced that the results are very different from Sarkhel's method.

Clarity:
The paper is well-organized and each section serves its purposes. There are nice definition such as the definition of single occurrence and extreme assignment.
However in some places there is difficulty understanding all the theorems without explicit examples. For instance, section 4.1. requires examples mentioned in the text not in the additional document. Without this example it is a bit difficult to understand the rest of this section.
Also in Algorithm2, the function getAllTautologyAtExtremes adds no new information and maybe isTautologyAtExtremes should have been explained in pseudo code instead.

Originality:
The originality of this paper is its major issue. First rule is essentially breaking the theory into sub-theories, assigning new weights according to their presence in an equivalence class.
It seems this approach is not that different than previous network simplifications such as the decomposer. According to the proof of Theorem 4.2 a decomposer has the same definition of a single occurrence of an equivalence class. It seems that the decomposition idea in this work is very similar to the decomposer idea, only at a different level (on the MAP inference applied over a MLN). While the assignment of the new weights is different and novel but as claimed by corollary 4.1 this assignment is to some extend intuitive.
Similarly the second rule is exactly derived from the paper of Sarkhel et. al. and thus is not considered a new innovative rule on its own.
I would argue that the idea in this work is a modification of previous rules on MLN networks (specifically Sarkhel et. al) now applied to the MAP queries. There is not enough motivation to why this work and its ideas are novel.

Significance:
According to the mentioned reasons this paper has minor significance. Due to the fact that its data are all from previous work or benchmarks, the results are not clear ( mentioned before) and the authors have failed to mention what happens to the cases where non of the formulas are single occurrences: there is a strong restriction on the theory. The major claim of independence is also mentioned in previous work and thus is not a new result for MLNs.
Summary: Work has strong mathematics and logical reasoning. However seems to be to some extend incremental to previous work, adding effectively one rule which is reducing the theory for the inference step.
Author Feedback
Author rebuttal: Reviewer 1 (Assigned_Reviewer_17): We will make the suggested changes to allow for lifted output format and deal with the existentials in the final version. In general, singleton formulas do not break any required properties. Our Friends and Smokers MLN has a singleton rule for each predicate (see Section 6.1) including Friends(X,Y). Once the tautologies at extremes are removed, X,Y variables in Friends(X,Y) predicate belong to different equivalence classes (in the remaining theory) and hence, single occurrence rule is still applicable. The sign of the singleton rule literal (negated or not) does not matter for the above argument to apply. As MLNs get more complex, we expect that not all the variables will be single occurrence, but our rules can still give good benefits. IE domain is a good example of this. Trying out other useful MLNs is a direction for future work. We can allow functions in the limited setting as described by the original MLN paper (Richardson and Domingos 2006) i.e. when they evaluate to constants already in the domain and are known in advance. We will make the correction on line 131. In line 312, it is necessary since it is a clausal formula and it must be a tautology at all extreme assignments. We are not sure of a direct connection with monotone formulas or safe negation. For formulas such as transitivity, all the groundings are true at any of the extreme assignments (including all 0's), but there are other "intermediate" assignments which lead to false groundings.

Reviewer 3 (Assigned_Reviewer_41): For the IE domain, one should compare figure 1(b) in our paper with figure 3(i) of Sarkhel et al. The graphs in the two papers for NSLGRB are consistent with each other, the only difference being that our run of NSLGRB reaches the optimal at 15 seconds (compared to about 65 seconds reported in their paper). We attribute this primarily to different processor speeds. Further, figure 1(b) (in our paper) clearly shows that SOLGRB is able to reach the optimal solution much faster than NSLGRB. For the Student network, Figure 2(b) in our supplement and figure 3(c) in Sarkhel et al. are again consistent with each other. Sarkhel et al. show a constant cost (of about 1 million) across time which is same as in our run. But a more careful analysis (figure 2(b) in our supplement) shows that this cost is in fact not the optimal one. NSLGRB never reaches the optimal cost of 250k unsat. clauses, which our algorithm can find in almost no time. For FS, figure 2(b) clearly shows a win for our approach (Sarkhel et al. do not report results for FS).

Sarkhel et al. plot multiple graphs, one for each domain size, showing the variation in cost with increasing time. The maximum domain size considered is 500. Unlike our paper, there is no single graph plotting time (or memory) with varying domain size. Further, the real scalability test comes after around size 500 (as shown by our figure 1(a) for IE or 2(a) for FS) where the two algorithms start to take different times to reach the optimal. For Student (figure 2(a) in the supplement), NSLGRB can't even reach the optimal beyond the domain size of 125. Our memory graphs (1(c), 2(c) in the main paper and 2(c) in the supplement) clearly show the benefit of SOLGRB over NSLGRB.

Figure 4 in Sarkhel et al. plots the optimal cost with varying domain size but it is only done upto size 100 which is very small and does not give much information about scalability of the algorithm for larger domain sizes (no such plot for IE has been given).

We will take into account the suggestions about including the example and the pseudocode in the final version (space permitting).

We believe there has been a misunderstanding with regards to Theorem 4.2. The theorem only claims that whenever decomposer is applicable, single occurrence is also applicable. The other way round is clearly not true. As in the proof of Theorem 4.2, in addition to being single occurrence, a decomposer also requires that the variable be present in every predicate in the formula which is very restrictive. Let us revisit the example in the text (Section 3.1 and Section 4.2):
Parent(X,Y) ^ Friend(Y,Z) => Knows(X,Z). Here, every variable is single occurrence but there are no decomposers. It's not difficult to come up with additional such examples. So, our single occurrence rule is definitely a novel contribution.

Similarly, we argue that our tautology at extremes (second) rule is a novel one. To the best of our knowledge, none of the previous works have come up with such a rule in any form. Sarkhel et al. introduce a notion of extreme assignments only in a very restricted setting of non-shared MLNs. Further, they have not used this idea to come up with a rule similar to ours. This is a very powerful rule enabling us to handle difficult formulas such as symmetry and transitivity which could not be dealt with earlier.

Our contribution also becomes significant given that there are only a few lifted inference rules in the literature where the key ones include Conditioning, Decomposition, Binomial rule, Lifted Decomposition, Sarkhel et al.'s rule and Partial Grounding (when nothing else applies). Our paper adds two additional rules (for MAP problem) which advances the state of the art substantially.

We have tried to perform a fair experimental analysis by considering domains (real as well as artificial) from similar existing work (IE, Student domains in Sarkhel et al.) as well as other existing MLN benchmarks (FS domain).

Our proposed rules already cover a large class of MLN theories which could not be efficiently handled earlier. For the domains that we experimented with, we always found single occurrence variables to reduce the theory on. When there are no more single occurrence variables, combining our rules with other existing lifting rules can lead to greater simplification of the theory as outlined in Section 7. Exploring this in detail is a part of the future work.